# AUTOCLIP: AUTO-TUNING ZERO-SHOT CLASSIFIERS FOR VISION-LANGUAGE MODELS

**Jan Hendrik Metzen, Piyapat Saranrittichai & Chaithanya Kumar Mummadi**
Bosch Center for Artificial Intelligence, Robert Bosch GmbH
`janhendrik.metzen@de.bosch.com`

## ABSTRACT

Classifiers built upon vision-language models such as CLIP have shown remarkable zero-shot performance across a broad range of image classification tasks. Prior work has studied different ways of automatically creating descriptor sets for every class based on prompt templates, ranging from manually engineered templates over templates obtained from a large language model to templates built from random words and characters. Up until now, deriving zero-shot classifiers from the respective encoded class descriptors has remained nearly unchanged, i.e., classify to the class that maximizes cosine similarity between its averaged encoded class descriptors and the image encoding. However, weighing all class descriptors equally can be suboptimal when certain descriptors match visual clues on a given image better than others. In this work, we propose AUTOCLIP, a method for *auto-tuning zero-shot classifiers*. AUTOCLIP tunes per-image weights to each prompt template at inference time, based on statistics of class descriptor-image similarities. AUTOCLIP is fully unsupervised, has very low computational overhead, and can be easily implemented in few lines of code. We show that AUTOCLIP outperforms baselines across a broad range of vision-language models, datasets, and prompt templates consistently and by up to 3 percent point accuracy.

## 1 INTRODUCTION

Classifiers built upon vision-language models (VLMs) such as CLIP (Radford et al., 2021) and CoCa (Yu et al., 2022) have shown strong zero-shot transfer capabilities across various tasks. Such zero-shot transfer is appealing since it allows for obtaining high-performing classifiers on novel domains without the overhead of data acquisition and labelling. However, it has been observed that prompt engineering plays a crucial role for obtaining strong zero-shot classifiers, that is: zero-shot classifiers derived from VLMs need to be constructed based on a set of prompt templates (parameterized by the class name) that cover potential variation of the domain. These prompt templates can be hand-designed (Radford et al., 2021), generated by a large-language model (Menon & Vondrick, 2022), or randomly generated (Roth et al., 2023).

Prompts can also be learned via test-time prompt tuning (TPT) (Shu et al., 2022; Zhao et al., 2023). This approach makes the zero-shot classifier adaptable to the datum of interest, which is possible by effectively leveraging the knowledge of the general-purpose VLM. Shu et al. (2022) tune prompts so that the predictive entropy for a single image is minimized, while Zhao et al. (2023) maximizes a CLIP reward. These prior TPT methods require the VLM's image encoder to process several augmentations for each image. Moreover, gradients with respect to the prompts require backpropagation through the VLM's text encoder, thereby substantially increasing the overall inference cost.

We propose to not tune the prompts but instead use a large set of predefined and fixed prompt templates and to adapt the weights of those prompt templates for each image at test-time. This approach has the major advantage that adaptation takes place entirely in the embedding space without requiring additional forward or backward passes through the VLM's encoders, which significantly lowers the test-time computation and memory overhead compared to prior TPT methods. Our work is similar to Allingham et al. (2023) with the main advantage that our approach can adapt weights based on single samples and without requiring access to the pre-training feature distribution.

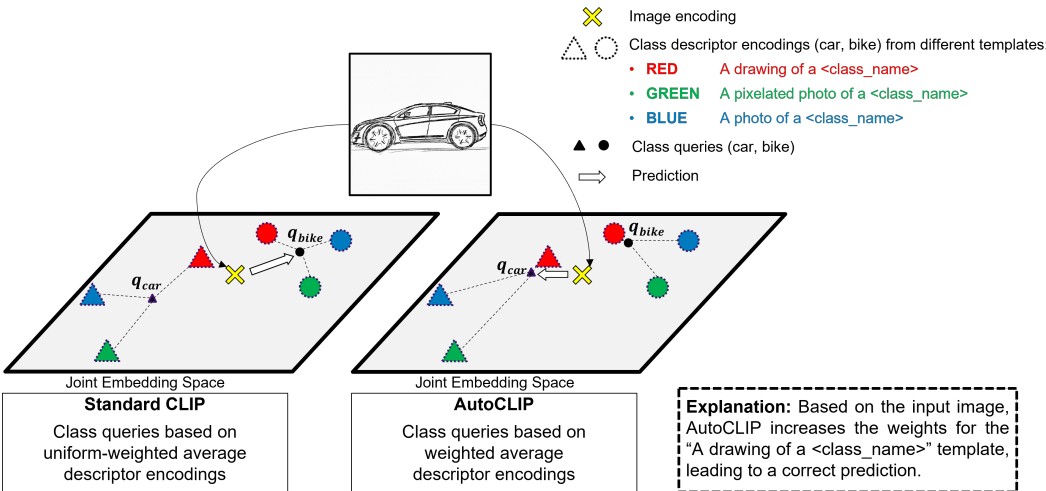

Figure 1: **Conceptual Illustration of AUTOCLIP.** CLIP's zero-shot classifiers are based on a set of prompt templates ("A photo of a <class_name>", "A drawing of a <class_name>", ...). Inserting class names into these templates gives a set of class descriptors that are encoded into a joint embedding space together with the respective image. Standard CLIP averages encoded class descriptors into class queries $q_c$, and classifies to the class that has maximal cosine similarity with the encoded image. However, this ignores that some prompt templates describe the image of interest better than others: for instance, when the image is a drawing, the template "A drawing of a <class_name>" results in stronger class descriptors than other templates and should thus be weighted higher when computing class queries. AUTOCLIP determines such weights directly from class descriptor-image similarities in the embedding space. Here, the car image is taken from Atkinson (2015).

We briefly summarize the standard way of constructing zero-shot classifiers from VLMs (see Figure 1 left). At first, a collection of prompt templates are instantiated for each class to form a set of class descriptors (e.g., "A photo of a *car*", and "A drawing of a *car*" are sample class descriptors of class *car*). These descriptors are processed by the text encoder and the resulting encoded descriptors are averaged to obtain the image-independent class queries (e.g. $q_{car}$). Besides, the image encoder processes the input image to be classified to get the image encoding, which lies in the same embedding space as class queries. The cosine similarity of the encoded image to every (averaged) class query is computed, and the output prediction is assigned to the class with maximum similarity.

This work follows a similar zero-shot classification setup, except that we change how class queries are computed. Instead of a simple average of the encoded class descriptors, we propose to take a weighted average, wherein weights of the encoded class descriptors are automatically tuned for each image separately. Weights are determined in a manner that prompt templates whose resulting class descriptors are closer to the respective image embedding get higher weightage than those being less similar (see Figure 1 right). Our approach is motivated by the intuition that prompt templates with high similarity describe relevant properties of the image better than ones with lower similarity (see Figure 6 for evidence supporting this intuition). We denote our method that automatically adapts weights of the encoded class descriptors for each image as AUTOCLIP.

We empirically show that AUTOCLIP improves the performance of zero-shot classifiers across many datasets, VLMs, and prompt strategies with little inference-time overhead. Note that AU-TOCLIP is fully zero-shot as it does not require any supervision from the target task. Furthermore, AUTOCLIP makes no assumptions on the underlying VLM and can thus be broadly applied, potentially also to multi-modal models beyond VLMs such as ImageBind (Girdhar et al., 2023b).

Overall, our main contributions are as follows: we introduce AUTOCLIP (Section 3.2), a novel procedure for constructing zero-shot classifiers from vision-language models. AUTOCLIP leverages statistics of class descriptor-image similarities to automatically determine weights of the prompt templates. We further discuss a method for automatically tuning AUTOCLIP's step size such that the entropy of the prompt template's weights is controlled (Section 3.4). We propose a default

entropy reduction factor, which is shared across all the experiments. By this, AUTOCLIP comes essentially without free hyperparameters, which is important as hyperparameters cannot be tuned in zero-shot settings. We evaluate AUTOCLIP on a large number of datasets, vision-language models, and prompt templates (Section 4). We find that it improves performance on the vast majority (85%) of settings, by 0.45 percent point accuracy on average, and by up to 3 percent point in some settings. This benefit comes essentially for free with the only cost being a very small inference time overhead.

## 2 RELATED WORK

**Vision-Language Pretraining.** Deep learning with vision-language pretraining has enabled zero-shot transfer capabilities, i.e., the resulting vision-language models (VLMs) are able to perform zero-shot classification on vastly diverse unseen target datasets given only text prompts of individual target classes. CLIP is one of the state-of-the-art VLMs pretrained on the well-curated WebImage-Text dataset containing 400 millions image-text pairs using a contrastive loss (Radford et al., 2021). In terms of datasets used, ALIGN requires less dataset preprocessing enabling training on a dataset of over a billion image-text pairs (Jia et al., 2021). Florence (Yuan et al., 2021) expands models to other common modalities (e.g., videos). In terms of the training loss, CoCa (Yu et al., 2022) leverages an additional captioning loss allowing models to be used in generative applications. In our work, we study how to optimally use text prompts of the target classes with these VLMs.

**Prompt Construction.** Conventionally, one or several manually designed text prompts per target class are employed for zero-shot classification (Radford et al., 2021; Jia et al., 2021). Recent research demonstrates that introducing additional prompts can improve overall performance. DCLIP (Menon & Vondrick, 2022) generates additional prompts based on querying the large-language model GPT-3 (Brown et al., 2020). WaffleCLIP (Roth et al., 2023) has shown that classification performance can be further boosted by appending random words or characters to predefined prompt templates. To derive a zero-shot classifier, these works weight all text prompts uniformly. In contrast, we propose an approach to adjust weights of individual prompts per input sample dynamically at test time.

**Test-Time Adaptation.** Our work can be considered as a test-time adaption approach for VLMs. TENT (Wang et al., 2020) demonstrates that adapting models to minimize prediction entropy can improve model performance at test time. In the context of VLMs, TPT (Shu et al., 2022) optimizes prompts of target classes based on the entropy minimization objective. RLCF (Zhao et al., 2023) demonstrates that minimizing the entropy objective can lead to overfitting under distribution shift and proposes adaptation based on average CLIP scores. In contrast to these previous works, we do not perform any adaptation of prompts or model parameters, but refine weights of individual (encoded) prompts, which is considerably cheaper in terms of computation and memory consumption. Most similar to us is Zero-shot Prompt Ensembling (ZPE) (Allingham et al., 2023), which also determine prompt weights in embedding space. However, ZPE requires an entire batch of target domain samples and the availability of image features that represent the feature distribution in pre-training ("source domain"). In contrast, our work operates on single images in a source-free setting.

## 3 AUTOCLIP

We outline the common approach for building zero-shot classifiers for VLMs like CLIP in Section 3.1. Thereupon, we detail our proposed AUTOCLIP as an auto-tuned alternative in Section 3.2, followed by describing how the required gradient can be calculated in closed-form in Section 3.3, and finally explain how AUTOCLIP's step size can be automatically determined in Section 3.4.

### 3.1 BACKGROUND: ZERO-SHOT CLASSIFIERS FOR VISION-LANGUAGE MODELS

Let us consider a classification task $\mathcal{X} \mapsto \mathcal{C}$, where $\mathcal{X}$ corresponds to the input domain and $\mathcal{C} = \{c_1, \ldots, c_C\}$ is a set of $C$ classes. We assume that there exists a pretrained VLM such as CLIP that provides a joint embedding space $\mathcal{E}$ and corresponding embedding functions $E_X : \mathcal{X} \mapsto \mathcal{E}$ that maps input data $x \in \mathcal{X}$ into embedding space $\mathcal{E}$ and $E_T : \mathcal{T} \mapsto \mathcal{E}$ that maps text into the same embedding space $\mathcal{E}$. Let there be $K$ prompt templates $t_1, \ldots t_K : \mathcal{C} \mapsto \mathcal{D}$ that map class name $c \in \mathcal{C}$ to (textual) class descriptors $d \in \mathcal{T}$. These prompt templates can be either manually designed (Radford et al., 2021), generated by a large language model (Menon & Vondrick, 2022), or randomly

---

**Algorithm 1** Zero-Shot Classifier

1: $d \leftarrow \{t_i(c_j) \,|\, i \in \{1, \ldots, K\}, j \in \{1, \ldots, C\}\}$      ▷ Generate $K \times C$ class descriptors
2: $e^{(x)} \leftarrow E_X(x)/||E_X(x)||_2$      ▷ Encode image of interest $x$ with VLM
3: $e_{ij}^{(d)} \leftarrow E_T(d_{ij})/||E_T(d_{ij})||_2$      ▷ Encode all class descriptors with VLM
4: $w_i \leftarrow 1/K$      ▷ Uniform prompt template weights
5: **for** $j \in 1, \ldots, C$ **do**
6:      $q_j \leftarrow \sum_{i=1}^{K} w_i e_{ij}^{(d)}$      ▷ Class queries as average of classes' descriptor encodings
7:      $s_j \leftarrow e^{(x)} \cdot q_j$      ▷ Compute cosine similarity between $e^{(x)}$ and class query $q_j$
8: **end for**
9: $j^\star \leftarrow \arg\max_j s_j$      ▷ Assign $x$ to class $c_{j^\star}$ with maximum similarity

---

generated (Roth et al., 2023). Algorithm 1 summarizes the standard zero-shot classifier for VLMs: average the class descriptor encodings $e^{(d)}$ into class queries $q_j$, then compute cosine similarities $s_j$ between class query and encoded image $e^{(x)}$, and classify to the class that maximizes similarity.

## 3.2 AUTO-TUNING ZERO-SHOT CLASSFIERS

AUTOCLIP modifies step (4) in Algorithm 1. Instead of computing class queries as simple average of class descriptor encodings $q_j = 1/K \sum_{i=1}^{K} e_{ij}^{(d)}$, AUTOCLIP uses a weighted average: $q_j = \sum_{i=1}^{K} w_i e_{ij}^{(d)}$ with learnable $w$ satisfying $w_i \geq 0$, $\sum_{i=1}^{K} w_i = 1$, which we enforce by reparametrizing $w = \mathrm{softmax}(\rho)$ and $\rho \in \mathbb{R}^K$. AUTOCLIP's guiding intuition (see Figure 1) is to assign higher weights $w_i$ to prompt templates $t_i$ that result in class descriptor encodings $e_{ij}^{(d)}$ that are more similar to the encoded image $e^{(x)}$, that is: $t_i$ with large $e_{ij}^{(xd)} = e_{ij}^{(d)} \cdot e^{(x)}$ $(j = 1, \ldots, C)$. This is inspired by the observation that class descriptors having higher similarity in the embedding space describe the image better (according to contrastive pretraining objectives in typical VLMs).

When determining the template's weights $w$, we have $C$ descriptor-image similarities $e_{ij}^{(xd)}$ for each template $t_i$. AutoCLIP needs to aggregate those $C$ similarities across classes when assigning larger weight to more relevant prompt templates. Intuitively, simply averaging all $C$ similarities ("mean" aggregation) ignores that, in the classification objective, we ultimately only care about classes that result in the descriptors closest to $e^{(x)}$; however, taking only the class with highest similarity per template into account ("max" aggregation) ignores inherent ambiguity in the image and was found to be suboptimal (Roth et al., 2023). We propose a middle ground of aggregating via a smooth approximation to the maximum function via $\mathrm{logsumexp}_j(e_{ij}^{(xd)}) = \log \sum_{j=1}^{C} \exp e_{ij}^{(xd)}$. This logsumexp aggregation takes all classes into account but assigns higher importance to more relevant classes (ones resulting in higher similarities to the image $x$). AUTOCLIP then determines weights $w_i$ such that $\mathrm{logsumexp}_j(s_j) = \mathrm{logsumexp}_j(\sum_{i=1}^{K} w_i e_{ij}^{(xd)}) = \mathrm{logsumexp}_j(\mathrm{softmax}(\rho) \cdot e_{:j}^{(xd)})$ gets increased by one step of gradient ascent in the direction of $\nabla_\rho \mathrm{logsumexp}_j(\mathrm{softmax}(\rho) \cdot e_{:j}^{(xd)})$. We note that $-\mathrm{logsumexp}$ has been interpreted as the energy function of a data point (for appropriately trained classifiers) (Grathwohl et al., 2020); in this view, AUTOCLIP can be interpreted as minimizing the energy and maximizing the probability density $p(x)$ of $x$ under the zero-shot classifier.

We summarize AUTOCLIP in Algorithm 2. We initialize $\rho = \mathbf{0}$, which corresponds to an unweighted average of the class descriptor encodings (Line 4). Similar to Algorithm 1, we compute the pairwise cosine similarities $s_j$ between encoded image $e^{(x)}$ and class queries $q_j$ (Line 5-8). Instead of directly classifying to the class with maximum similarity to the image, AUTOCLIP updates the class descriptor weights first. For this, the gradient $g = \nabla_\rho \mathrm{logsumexp}_j(s_j)$ is computed (Line 9), an appropriate step size $\alpha$ is selected (Line 10, see Section 3.4), and $\rho = \alpha \cdot g$ and $w = \mathrm{softmax}(\rho)$ are updated (Line 11). Based on the new $w$, AUTOCLIP computes updated class queries $q_j$ and class-image similarities (Line 12-15) and finally selects the class with maximum similarity for the image (Line 16). We note that AUTOCLIP is permutation-invariant in the prompt templates $t_i$.

We note that Line 5-11 could be repeated for several iterations with smaller step sizes; however preliminary experiments indicate no advantage of doing more than one iteration. We call AUTOCLIP

---

**Algorithm 2** AUTOCLIP: Auto-Tuned Zero-Shot Classifier

1: $d \leftarrow \{t_i(c_j) \,|\, i \in \{1, \ldots, K\}, j \in \{1, \ldots, C\}\}$      $\triangleright$ Generate $K \times C$ class descriptors
2: $e^{(x)} \leftarrow E_X(x)/||E_X(x)||_2$      $\triangleright$ Encode image of interest $x$ with VLM
3: $e_{ij}^{(d)} \leftarrow E_T(d_{ij})/||E_T(d_{ij})||_2$      $\triangleright$ Encode all class descriptors with VLM
4: $\rho \leftarrow \mathbf{0}; \quad w_i \leftarrow \mathrm{softmax}(\rho)$      $\triangleright$ Uniform weights $w_i = 1/K$
5: **for** $j \in 1, \ldots, C$ **do**
6:      $q_j \leftarrow \sum_{i=1}^{K} w_i e_{ij}^{(d)}$      $\triangleright$ Class queries as weighted average of classes' descriptor encodings
7:      $s_j \leftarrow e^{(x)} \cdot q_j$      $\triangleright$ Compute cosine similarity between $e^{(x)}$ and class query $q_j$
8: **end for**
9: $g \leftarrow \nabla_\rho \log \sum_{j=1}^{C} \exp(s_j)$      $\triangleright$ Compute gradient (Section 3.3)
10: $\alpha \leftarrow \mathrm{BISECT}(\mathrm{softmax\_entropy}(\alpha \cdot g) - \beta \log_2 K, 0, 10^{10})$ $\triangleright$ Determine stepsize (Section 3.4)
11: $\rho \leftarrow \alpha \cdot g; \quad w_i \leftarrow \mathrm{softmax}(\rho)$      $\triangleright$ Update $\rho$ with one gradient ascent step and step size $\alpha$
12: **for** $j \in 1, \ldots, C$ **do**
13:      $q_j \leftarrow \sum_{i=1}^{K} w_i e_{ij}^{(d)}$      $\triangleright$ Class queries as weighted average of classes' descriptor encodings
14:      $s_j \leftarrow e^{(x)} \cdot q_j$      $\triangleright$ Compute cosine similarity between $e^{(x)}$ and class query $q_j$
15: **end for**
16: $j^\star \leftarrow \arg\max_j s_j$      $\triangleright$ Assign $x$ to class $c_{j^\star}$ with maximum similarity

---

"auto-tuned" because its weights $w$ are automatically adapted for every input independently. Moreover, we note that in practice, models like CLIP scale $e^{(xd)}$ by a learned temperature (exponential logit scale) $\tau$ to obtain well calibrated classifiers; we use the same temperature for scaling $e^{(xd)}$ in the logsumexp aggregation (as there is no labelled data in a zero-shot setting on which a temperature could be tuned).

### 3.3 CLOSED-FORM COMPUTATION OF GRADIENT

While $\nabla_\rho \mathrm{logsumexp}(s)$ can be easily computed using automatic differentiation, we note that there can be runtime environments for inference such as on edge devices where running automatic differentiation is undesirable. For such cases, the gradient $\nabla_\rho \mathrm{logsumexp}_j(s_j)$ can also be computed in closed-form: $\left(\nabla_\rho \mathrm{logsumexp}_j(s_j)\right)_i = \sum_{k=1}^{K}(\sum_{j=1}^{C} \mathrm{softmax}(s)_j \cdot e_{ij}^{(xd)}) \cdot w_i(\delta_{ik} - w_k)$, with $\delta_{ij}$ being the Kronecker delta function with $\delta_{ii} = 1$ and $\delta_{ij} = 0$ for $i \neq j$.

### 3.4 AUTO-TUNING THE STEP SIZE

The only free hyperparameter of AUTOCLIP is the step size $\alpha$. We note that in a zero-shot setting, there is by definition no labeled data on which such free hyperparameters can be tuned. Because of this, free hyperparameters need to be selected globally in a dataset-independent manner. However, a global choice for the step size $\alpha$ is problematic since the scale of the gradient $g = \nabla_\rho \mathrm{logsumexp}(s)$ depends on the dataset, and the step size would have to be adapted accordingly. We address this by proposing a different parameterization in which the free hyperparameter is easily interpreted and the step size $\alpha$ is a derived quantity. Specifically, we control the entropy of the query weights $w$, $\mathrm{entropy}(w) = -\sum_{i=1}^{K} w_i \log_2 w_i$. The standard, uniform weights have maximum entropy $\log_2 K$ and we set the target entropy to $\beta \cdot \log_2 K$, where the entropy reduction factor $\beta \in [0, 1]$ is the new free hyperparameter that we set globally to $\beta = 0.85$. Intuitively, $\beta \to 1$ corresponds to more equally weighted prompt templates while $\beta \to 0$ to selecting the prompt template with maximum similarity. We present an ablation of the effect of $\beta$'s choice on AUTOCLIP in Figure 4.

With $\mathrm{softmax\_entropy}(\alpha \cdot g)$ denoting the entropy of the weights $w = \mathrm{softmax}(\alpha \cdot g)$, selecting the step size $\alpha$ is now equivalent to solving for $f(\alpha) = 0$ for $f(\alpha) = \mathrm{softmax\_entropy}(\alpha \cdot g) - \beta \cdot \log_2 K$. As $\mathrm{softmax\_entropy}(\alpha \cdot g)$ monotonically decreases with $\alpha$, we use bisection on $\alpha \in [0, 10^{10}]$ for finding $\alpha$ with $f(\alpha) \approx 0$. We note that $\mathrm{softmax\_entropy}(0 \cdot g) = \log_2 K$ and thus $f(0) > 0$ for all $\beta < 1$; similarly, $\mathrm{softmax\_entropy}(\alpha \cdot g) \approx 0$ for $\alpha = 10^{10}$ in all settings we considered and thus $f(10^{10}) < 0$ for all $\beta > 0$, which together satisfies the prerequisites for running bisection. The additional bisection has little overhead compared to the cost of encoding the image $x$ with $E_x$ (see Section A.1 in the appendix for details).

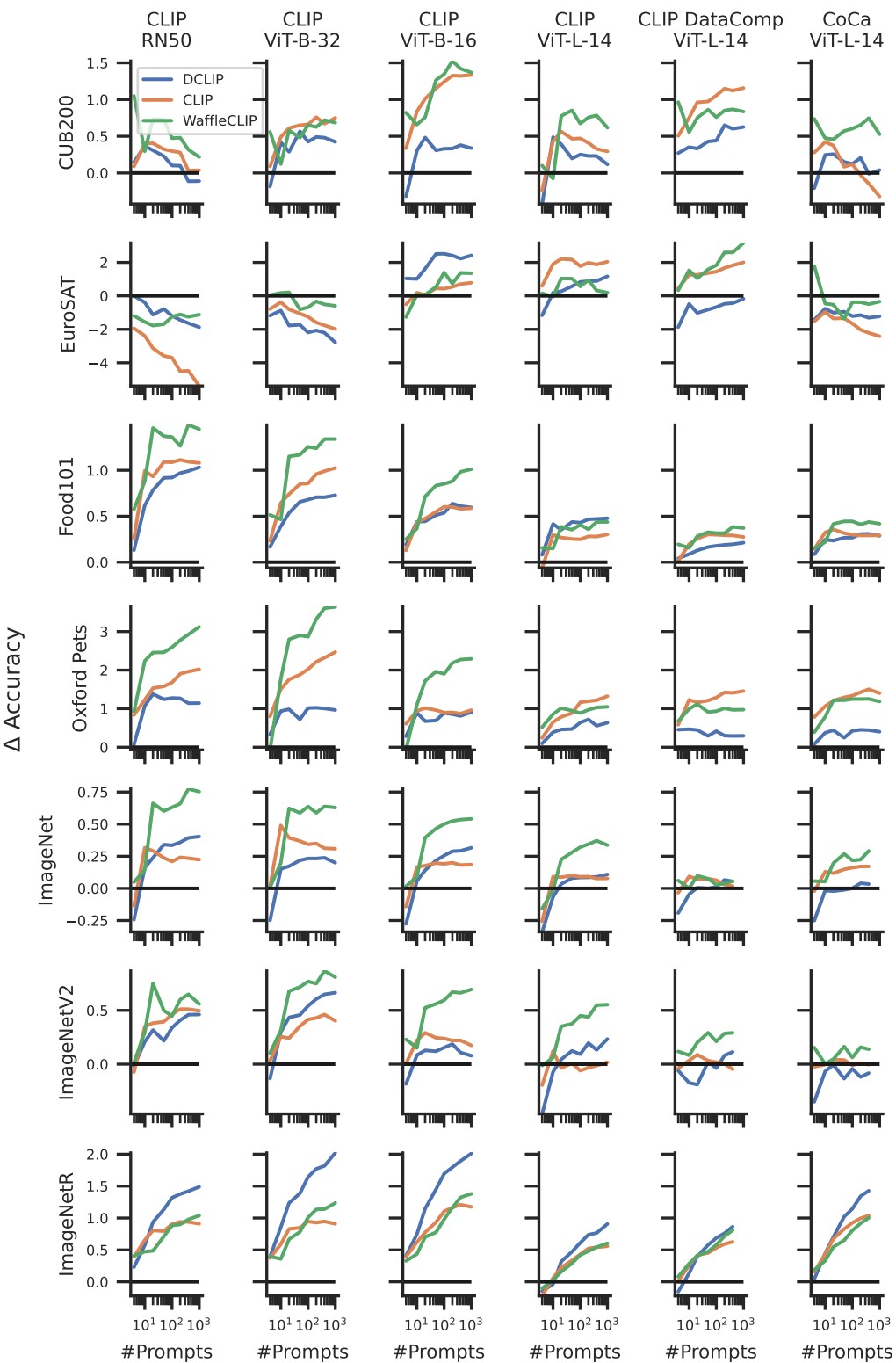

Figure 2: Accuracy improvement (Δ Accuracy) of AUTOCLIP over baseline zero-shot classifier across models, datasets, and prompt ensembles, averaged over 7 runs.

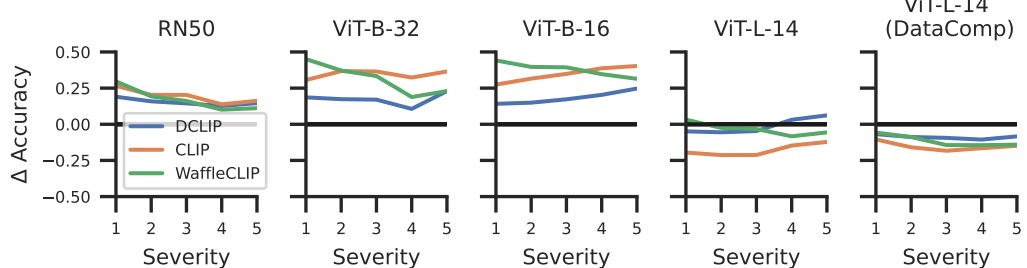

Figure 3: ImageNet-C accuracy improvement ($\Delta$ Accuracy) of AUTOCLIP over baseline zero-shot classifier for $K = 100$ across models, corruption severity and prompt ensembles, averaged over corruptions and 7 runs.

|  | CLIP RN50 | CLIP ViT-B-32 | CLIP ViT-B-16 | CLIP ViT-L-14 | DataComp ViT-L-14 | CoCa ViT-L-14 |
|---|---|---|---|---|---|---|
| CUB200 | 47.75 (+0.5) | 52.84 (+0.7) | 57.12 (+1.3) | 64.43 (+0.7) | 84.79 (+0.8) | 73.90 (+0.6) |
| EuroSAT | 34.95 (-1.2) | 46.16 (-0.7) | 55.93 (+1.4) | 55.09 (+0.6) | 65.09 (+1.8) | 54.77 (-0.4) |
| Food101 | 80.26 (+1.4) | 84.13 (+1.3) | 88.85 (+0.9) | 93.71 (+0.4) | 94.52 (+0.3) | 90.46 (+0.4) |
| Oxford Pets | 83.09 (+2.6) | 85.63 (+2.9) | 85.89 (+1.9) | 91.64 (+0.9) | 92.82 (+0.9) | 92.03 (+1.2) |
| ImageNet | 60.42 (+0.6) | 63.80 (+0.6) | 68.70 (+0.5) | 75.89 (+0.3) | 79.07 (+0.0) | 75.63 (+0.2) |
| ImageNetV2 | 53.44 (+0.4) | 56.49 (+0.8) | 62.54 (+0.6) | 70.17 (+0.4) | 72.21 (+0.2) | 68.08 (+0.1) |
| ImageNetR | 29.32 (+0.9) | 51.04 (+1.0) | 59.13 (+1.0) | 73.98 (+0.4) | 78.85 (+0.6) | 75.59 (+0.8) |

Table 1: Accuracy of AUTOCLIP (and $\Delta$ Accuracy to baseline zero-shot classifier in parenthesis) for $K = 100$ WaffleCLIP prompt templates across models and datasets, averaged over 7 runs.

## 4 EXPERIMENTS

**Experimental Setting** In this section, we compare AUTOCLIP to standard zero-shot classifiers on a wide range of zero-shot image classification benchmarks and a variety of settings. We conduct experiments on the datasets CUB200 (Welinder et al., 2010), EuroSAT (Helber et al., 2019), Food101 (Bossard et al., 2014), Oxford Pets (Parkhi et al., 2012), ImageNet (Russakovsky et al., 2015), ImageNetV2 (Kornblith et al., 2019), ImageNet-R (Hendrycks et al., 2021), and ImageNet-C (Hendrycks & Dietterich, 2019). We study six different vision-language models: from CLIP (Radford et al., 2021), we use ResNet-50 (RN50) (He et al., 2015) and vision transformer (ViT-B/32, ViT-B/16, and ViT-L/14) model variants (Dosovitskiy et al., 2021). Moreover, we use the ViT-L/14 model variant from DataComp (Gadre et al., 2023) and the one trained with CoCa (Yu et al., 2022).

Additionally, we study three ways of generating prompt templates: 1) using the 80 manually designed templates from Radford et al. (2021) (CLIP), 2) templates based on querying a large-language model (DCLIP) (Menon & Vondrick, 2022), and 3) templates that append random words or characters to predefined prompt templates (WaffleCLIP) (Roth et al., 2023). We vary the number of templates from $K = 4$ to $K = 500$; if there is a fixed number of templates available such as in CLIP/DCLIP, templates are sampled with replacement. To account for randomness in the template construction/sampling, we report results averaged over 7 runs. We base our implementation on `https://github.com/ExplainableML/WaffleCLIP` from Roth et al. (2023) and highly appreciate their code release under a permissible license. We report the difference of accuracy of AUTOCLIP compared to the baseline zero-shot classifier with uniform prompt template weights ("$\Delta$ Accuracy"). Absolute performance across different datasets and VLMs is shown in Table 1 (and in Table 2 and Table 3 in the appendix).

**Results** We present the main results in Figure 2. Overall, the figure contains 990 different combinations comparing AUTOCLIP with the baseline; AUTOCLIP is better in 840 cases ($\approx 85\%$) and on average it is better by 0.45 percent point accuracy. We also observe a trend that for larger number of prompt templates $K$, the advantage of AUTOCLIP ($\Delta$ Accuracy averaged across datasets, models

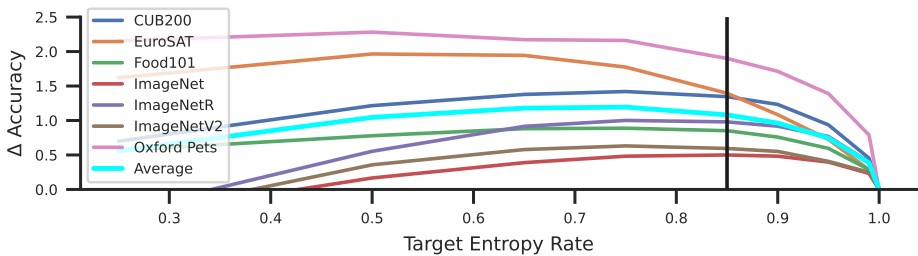

Figure 4: Ablation on target entropy rate $\beta$. Shown is the accuracy improvement ($\Delta$ Accuracy) of AUTOCLIP over baseline zero-shot classifier for a CLIP ViT-B-16, and 100 WaffleCLIP prompt templates, averaged over 7 runs.

and CLIP/DCLIP/WaffleCLIP) increases: from $\Delta = 0.06$ for $K = 4$ over $\Delta = 0.33$ for $K = 10$ and $\Delta = 0.49$ for $K = 50$ to $\Delta = 0.57$ for $K = 200$. When aggregating over models, datasets and number of prompt templates, AUTOCLIP achieves the largest average improvement for WaffleCLIP ($\Delta = 0.61$), but still improves for CLIP ($\Delta = 0.40$) and DCLIP ($\Delta = 0.29$). Taken together, the findings indicate that AUTOCLIP benefits from larger (increased K) and more diverse (WaffleCLIP) sets of prompt templates.

When comparing different vision-language models, AUTOCLIP brings the biggest benefit for CLIP ViT-B-16 ($\Delta = 0.68$) and the smallest one for CoCa ViT-L-14 ($\Delta = 0.19$), with all other models having average $\Delta$ between 0.36 and 0.52. Comparing different datasets, AUTOCLIP performs strongest on Oxford Pets ($\Delta = 1.15$) and worst on EuroSAT ($\Delta = -0.24$). We note that EuroSAT is the only setting on which AUTOCLIP hurts performance on average; on all other datasets, AUTOCLIP improves performance: $\Delta(\text{CUB200}) = 0.5$, $\Delta(\text{Food101}) = 0.52$, $\Delta(\text{ImageNet}) = 0.17$, $\Delta(\text{ImageNetV2}) = 0.2$, and $\Delta(\text{ImageNetR}) = 0.71$.

In Figure 3, we present results on ImageNet-C for WaffleCLIP with $K = 100$ for different severities and averaged across corruptions. AUTOCLIP consistently improves performance for the smaller vision-language models (RN50, ViT-B-32, ViT-B-16) and sees a minor drop of performance for the two ViT-L-14 variants. Averaged across all models, corruptions, and severities, AUTOCLIP improves performance by $\Delta = 0.11$. We provide plots for each corruption separately for WaffleCLIP prompt templates in the appendix in Figure 8. The biggest average benefit of AUTOCLIP is obtained for the low-frequency corruptions "saturate" ($\Delta = 0.22$), "brightness" ($\Delta = 0.22$), and "contrast" ($\Delta = 0.23$); the smallest average benefit for "shot-noise" ($\Delta = 0.05$) and "snow" ($\Delta = 0.06$).

**Ablations** We ablate AUTOCLIP's choice of the target entropy rate $\beta$ (which defaults to 0.85) and the objective function (defaults to logsumexp). In Figure 4, we observe that AUTOCLIP's performance for most datasets does not depend strongly on the specific choice of the target entropy rate $\beta$ as $\Delta$ Accuracy stays relatively constant in the range $\beta \in [0.7, 0.9]$. This is a desirable property as in a zero-shot setting without labeled data, tuning $\beta$ per dataset would be infeasible. For two datasets (Oxfort Pets and EuroSAT), our default value of $\beta = 0.85$ was suboptimal and a considerably smaller choice of $\beta = 0.7$ would have obtained considerably better results. Also on average, $\beta = 0.7$ performs favorably and we recommend this choice for future work on other datasets and tasks.

We motivated the choice of logsumexp as AUTOCLIP's aggregation/objective function in Section 3.2 as striking a good compromise between max and mean aggregation. In Figure 5, we empirically confirm that the logsumexp aggregation performs favorably compared to max/mean aggregation on all datasets. Moreover, it also outperforms entropy aggregation, which is a popular choice for test-time adaption (Wang et al., 2020; Shu et al., 2022).

In Figure 6, we show the prompt template weights ($K = 30$) obtained by AUTOCLIP on 500 Food101 samples. Samples are structured in 10 blocks of 50 samples each, where each block corresponds to one class. Prompt template weights are relatively similar for instances belonging to the same (unknown) class but vary substantially across classes. Some templates like the ones starting with "A tattoo of..." or "A origami of..." get consistently low weights as the images of the Food101 dataset do not look like tattoos or origami, while templates starting with "A photo of..." tend to get

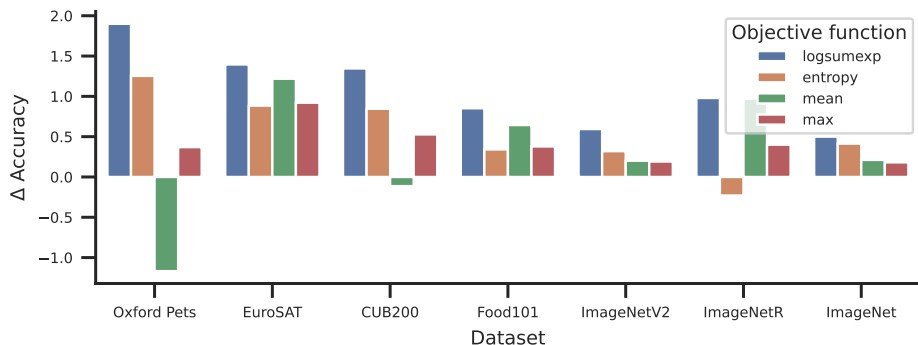

Figure 5: Comparison of different objective functions for auto-tuning. Shown is the accuracy improvement (Δ Accuracy) of AUTOCLIP over baseline zero-shot classifier for a ViT-B-16, and 100 WaffleCLIP prompt templates, averaged over 7 runs.

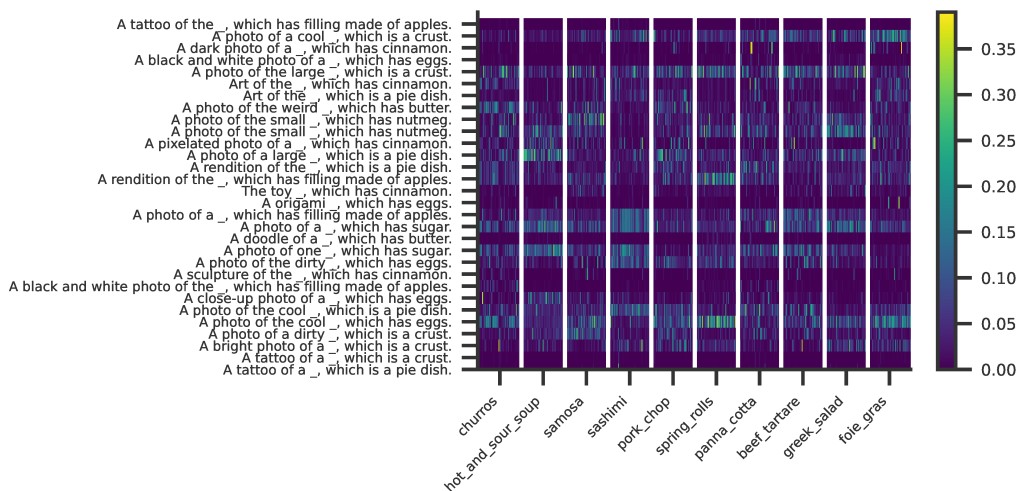

Figure 6: Illustration of prompt template weights $w$ on 500 samples from the Food101 dataset, with blocks of 50 samples belonging to the same (unknown) class. CLIP backbone is a ViT-B-16 and 30 DCLIP prompt templates are used.

higher weights, as Food101 contains mostly actual photos. Note that the weight distribution looks different on other datasets like ImageNet-R, with higher weights for "artistic" prompts (see Figure 7 in the appendix). Overall, this confirms that AUTOCLIP can adapt the zero-shot classifier on the fly to properties of the respective image.

## 5    CONCLUSION

We have proposed AutoCLIP, a method for improving zero-shot classifiers on vision-language models. AutoCLIP automatically tunes per-image weights of prompt templates before aggregating them into class queries. AutoCLIP improves performance over standard zero-shot classifiers on the vast majority of settings, with only minimal inference-time overhead. We believe that due to its simplicity and low cost, AutoCLIP has the potential to be broadly applied in conjunction with vision-language models. For future work, it will be exciting to explore if AutoCLIP can also benefit other zero-shot tasks built on top of multi-modal modals such as object detection with OWL-ViT (Minderer et al., 2022) or multi-modal prompting with ImageBind (Girdhar et al., 2023a).

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

# A  APPENDIX

## A.1  INFERENCE TIME OVERHEAD OF AUTOCLIP

In this paragraph, we provide some measurements on inference time overhead by AUTOCLIP. We provide numbers for the case of a ViT-L-14 on the Oxford Pets dataset. Here, encoding an image takes 12.64ms on a V100 (minimum over 100 images). The baseline "averaging" zero-shot classifiers takes additional 0.08ms (average over 640 samples) on top to classify a sample. AUTOCLIP takes additional 1.54ms (average over 640 samples) for classification when running bisection for autotuning the step size. For a fixed step size, the overhead of AUTOCLIP is 0.45ms. Thus, AUTO-CLIP with autotuning raises inference time from 12.64ms to 14.18ms. In contrast, TPT (Shu et al., 2022) and RLCF (Zhao et al., 2023), which did not report compute or memory requirements, require encoding multiple image augmentations. TPT states "We augment a single test image 63 times using random resized crops and construct a batch of 64 images, including the original one.", which means that the image encoding time (for instance the 12.64ms from above) is increased by a factor of 64x, plus additional overhead for backpropagating through the text encoder, which likely brings the inference time per sample close to 1s (or more if multiple test-time adaptation steps are conducted). We note that for bisection, we use an independent call to `scipy.optimize.bisect` (Virtanen et al., 2020) (maxiter=100, xtol=1e-2, rtol=1e-2). A batched variant of bisection could speed-up many workloads.

## A.2  ADDITIONAL EXPERIMENTAL RESULTS

We present additional experimental results. Table 2 and Table 3 show the absolute performance of AUTOCLIP on different datasets and VLMs for DCLIP and CLIP prompt templates, respectively, similar to Table 1 in the main paper for WaffleCLIP templates. Figure 7 illustrates prompt weights on the ImageNetR dataset. Figure 8 contains results of AUTOCLIP in terms of $\Delta$ Accuracy on ImageNetC for every corruption seperately.

In Figure 9, we show an additional comparison of AUTOCLIP to a stronger baseline which is based on TopR aggregation. In this TopR aggregation, for each image $R$ prompt templates are selected whose resulting encoded class descriptors have maximum average cosine similarity to the encoded image. We note that choosing R is non-trivial in a zero-shot setting due to the lack of labelled validation data. In the figure, we compare AUTOCLIP against this TopR-CLIP for $K = 100$ DCLIP prompt template, across the same VLMs and datasets as in Figure 2. We provide results for different choices of $R$: overall, for the best choice of $R = 20$, AUTOCLIP is better on $86\%$ of the cases and by $0.40$ percent point accuracy on average.

|  | CLIP RN50 | CLIP ViT-B-32 | CLIP ViT-B-16 | CLIP ViT-L-14 | DataComp ViT-L-14 | CoCa ViT-L-14 |
|---|---|---|---|---|---|---|
| CUB200 | 47.75 (+0.1) | 53.00 (+0.4) | 57.82 (+0.3) | 64.57 (+0.3) | 85.38 (+0.4) | 73.69 (+0.1) |
| EuroSAT | 36.39 (-1.2) | 45.88 (-2.2) | 59.22 (+2.5) | 57.89 (+0.8) | 60.08 (-0.7) | 57.15 (-1.2) |
| Food101 | 79.12 (+0.9) | 83.43 (+0.7) | 88.53 (+0.5) | 93.14 (+0.4) | 93.89 (+0.2) | 89.77 (+0.3) |
| Oxford Pets | 85.92 (+1.3) | 87.11 (+1.0) | 88.53 (+0.9) | 94.08 (+0.6) | 94.00 (+0.4) | 93.54 (+0.4) |
| ImageNet | 60.62 (+0.3) | 63.89 (+0.2) | 69.10 (+0.3) | 75.92 (+0.1) | 79.02 (+0.0) | 75.41 (+0.0) |
| ImageNetV2 | 53.60 (+0.3) | 56.73 (+0.5) | 62.22 (+0.2) | 70.01 (+0.1) | 71.95 (-0.0) | 67.91 (-0.0) |
| ImageNetR | 28.14 (+1.3) | 49.51 (+1.6) | 58.37 (+1.7) | 73.12 (+0.6) | 78.06 (+0.7) | 73.73 (+1.1) |

Table 2: Accuracy of AUTOCLIP (and $\Delta$ Accuracy to baseline zero-shot classifier in parenthesis) for $K = 100$ DCLIP prompt templates across models and datasets, averaged over 7 runs.

| | CLIP RN50 | CLIP ViT-B-32 | CLIP ViT-B-16 | CLIP ViT-L-14 | DataComp ViT-L-14 | CoCa ViT-L-14 |
|---|---|---|---|---|---|---|
| CUB200 | 47.00 (+0.3) | 52.36 (+0.7) | 56.99 (+1.2) | 63.94 (+0.5) | 85.52 (+1.1) | 73.99 (+0.1) |
| EuroSAT | 32.28 (-3.7) | 44.78 (-1.3) | 56.76 (+0.4) | 52.96 (+1.8) | 61.94 (+1.4) | 51.58 (-1.7) |
| Food101 | 79.69 (+1.1) | 83.64 (+0.9) | 88.83 (+0.6) | 93.33 (+0.2) | 94.55 (+0.3) | 90.36 (+0.3) |
| Oxford Pets | 84.30 (+1.7) | 85.20 (+2.0) | 88.42 (+0.9) | 93.24 (+1.2) | 93.79 (+1.3) | 92.67 (+1.3) |
| ImageNet | 59.90 (+0.2) | 63.31 (+0.3) | 68.43 (+0.2) | 75.38 (+0.1) | 79.29 (+0.1) | 75.79 (+0.2) |
| ImageNetV2 | 52.98 (+0.5) | 56.00 (+0.4) | 62.12 (+0.2) | 69.56 (-0.1) | 72.09 (+0.0) | 67.90 (-0.0) |
| ImageNetR | 27.11 (+0.9) | 47.74 (+0.9) | 56.28 (+1.1) | 71.30 (+0.4) | 78.26 (+0.5) | 74.51 (+0.9) |

Table 3: Accuracy of AUTOCLIP (and $\Delta$ Accuracy to baseline zero-shot classifier in parenthesis) for $K = 100$ CLIP prompt templates across models and datasets, averaged over 7 runs.

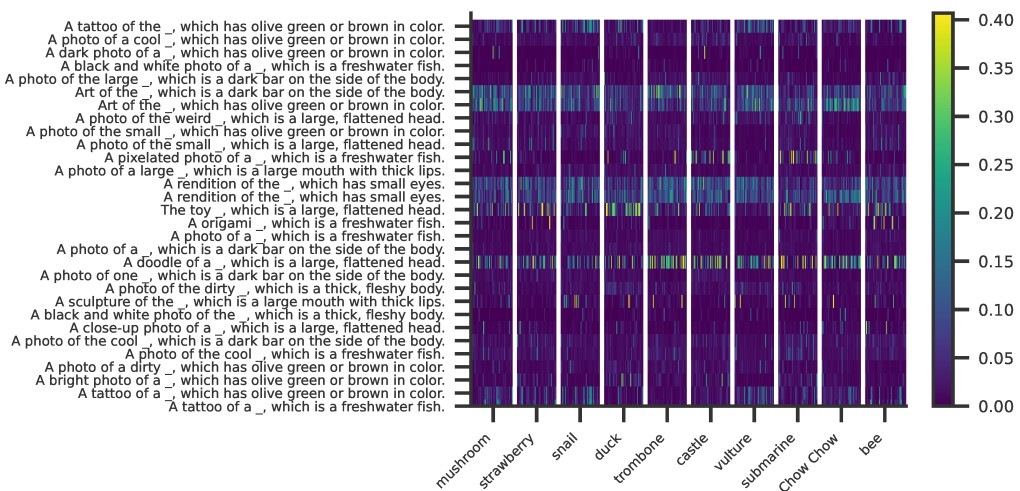

Figure 7: Illustration of prompt template weights $w$ on 500 samples from the ImageNetR dataset, with blocks of 50 samples belonging to the same (unknown) class. CLIP backbone is a ViT-B-16 and 30 DCLIP prompt templates are used.

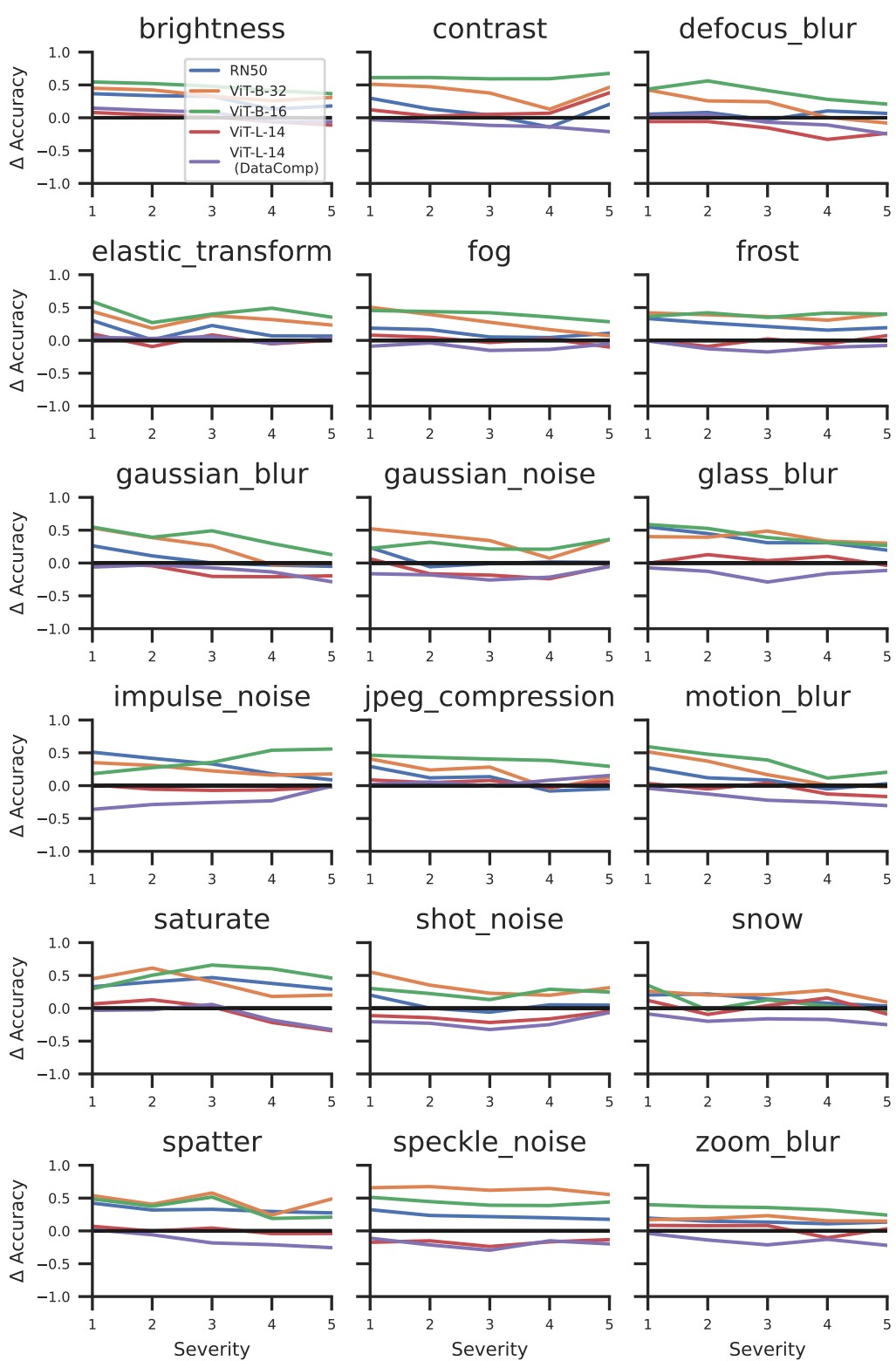

Figure 8: ImageNetC Accuracy improvement (Δ Accuracy) of AUTOCLIP over baseline zero-shot classifier for WaffleCLIP across models, corruptions, averaged over 7 runs.

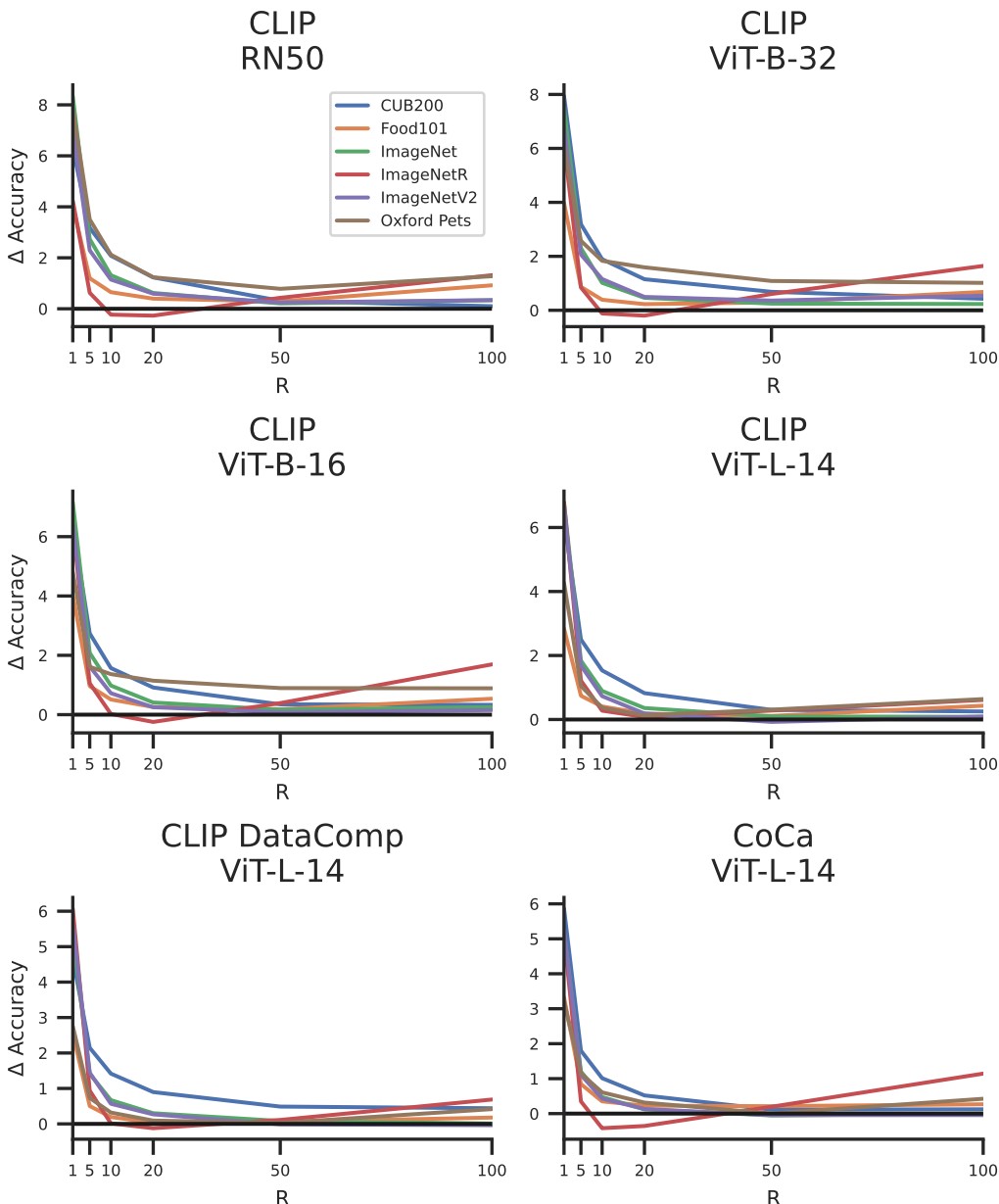

Figure 9: Accuracy improvement ($\Delta$ Accuracy) of AUTOCLIP with $K = 100$ DCLIP prompt templates over TopR zero-shot classifier with different values of $R$ across models, averaged over datasets and 7 runs.

