# OpenReview forum: "AutoCLIP: Auto-tuning Zero-Shot Classifiers for Vision-Language Models"
_ICLR.cc/2024/Conference — Submitted to ICLR 2024_

### Official Review · Reviewer_rfek · 2023-10-31

**Soundness:** 3 good
**Presentation:** 3 good
**Contribution:** 2 fair
**Rating:** 6
**Confidence:** 4

**Summary:**

This paper AutoCLIP, a method to adapt zero-shot classification inference in CLIP-like vision language models. As in CLIP, zero-shot classification is achieved by using text templates with class names to get the language embedding. Templates are predefined and can have multiple variants for the same class. When there are multiple templates for a class, CLIP averages them and then compare it with the image embedding. The authors argue that some templates can more accurately describe the images while some are not that good. So simply averaging them can be sub-optimal. To address this issue, AutoCLIP computes weighted average of the templates rather than simple average. Larger weights are assigned to templates that are more similar to the image embeddings. To this end, AutoCLIP computes logsumexp on the similarity s. Then, gradient g is computed from it wrt p in ascent direction, where weights w=softmax(p). p is then updated by the gradient and a step size. In experiments, this update is only conducted once as there is no advantage of more iterations. AutoCLIP is only used for the zero-shot classification inference and only for the procedure of computing language and image similarities, so there is no additional training beyond CLIP. AutoCLIP is verified by zero-shot classification on multiple image classification datasets.

**Strengths:**

1. The idea and motivation of AutoCLIP make sense. It is a reasonable way to improve the similarity calculation during zero-shot classification.
2. AutoCLIP does not require additional training beyond the vision and language model. Therefore, it is easy to apply AutoCLIP to existing zero-shot classifiers for improving accuracy.
3. The experiments are thoroughly conducted covering many classification datasets and ranging from CLIP to CoCa.
4. Experimental results suggest AutoCLIP is an effective approach to improve zero-shot classification accuracy. The improvements are about 1%; in most cases less than 1%.
5. The paper presentation is clear for audience.

**Weaknesses:**

1. The is one limitation: Current experiments suggest AutoCLIP can only be used for zero-shot classification, which is only one useful aspect of large vision and language model. Large vision and language model like CLIP is not about just doing zero-shot classification. For real applications, the impact of these models also lies in downstream tasks. For example, finetuning from CLIP pretrained parameters or using CLIP to directly assist the downstream tasks. It's better to explain some zero-shot classification applications.
2. In view of the above limitations, another concern comes out about the result. If a model has 1% improvement on ImageNet, it may bring more improvements when the pre-trained model is applied in downstream task. In case AutoCLIP is not intended for downstream task, the 1% improvement may look not significant. What this 1% improvement can bring?
3. The experiment in fig.3 is not clear. What does this experiment verify for? Any explain why ViT-L is worse? Does it mean AutoCLIP cannot well be scaled to larger models?
4. It is highly suggested to show evidence for "This is inspired by the observation that class descriptors having higher similarity in the embedding space describe the image better". This claim support the rationale of the proposed AutoCLIP. To my understanding, if this claim does not hold, then AutoCLIP does not hold.

**Questions:**

Is there any speed comparison? How much more inference time does the AutoCLIP need?

---

> ### Author Response · Authors · 2023-11-15
> **Response to Reviewer rfek**
>
> We thank the reviewer for the constructive review and helpful feedback. We appreciate that the reviewer finds that "The idea and motivation of AutoCLIP make sense", "it is easy to apply AutoCLIP to existing zero-shot classifiers for improving accuracy", "the experiments are thoroughly conducted covering many classification datasets and ranging from CLIP to CoCa.", and "the paper presentation is clear for audience.". We provide responses to the reviewer's questions and critical remarks below:
>
> > The is one limitation: Current experiments suggest AutoCLIP can only be used for zero-shot classification, which is only one useful aspect of large vision and language model. Large vision and language model like CLIP is not about just doing zero-shot classification. For real applications, the impact of these models also lies in downstream tasks. For example, finetuning from CLIP pretrained parameters or using CLIP to directly assist the downstream tasks. It's better to explain some zero-shot classification applications. In view of the above limitations, another concern comes out about the result. If a model has 1\% improvement on ImageNet, it may bring more improvements when the pre-trained model is applied in downstream task. In case AutoCLIP is not intended for downstream task, the 1\% improvement may look not significant. What this 1\% improvement can bring?
>
> In this work, we do focus on zero-shot classification. We are aware that this is only one potential usage of VLMs. However, it is an important enough use-case that several prior works have focused on it, such as for instance DCLIP (Menon
> & Vondrick, 2022) and WaffleCLIP (Roth et al., 2023). We also think that an improvement by 1 percent point accuracy is significant, particularly given that it comes without any retraining or custom prompt engineering by a pure change in the way the zero-shot classifier is constructed. However, we will further motivate zero-shot classification applications in the introduction as proposed by the reviewer.
>
>
> > The experiment in fig.3 is not clear. What does this experiment verify for? Any explain why ViT-L is worse? Does it mean AutoCLIP cannot well be scaled to larger models?
>
> Figure 2 provides ample evidence that AutoCLIP also brings benefits for larger models (ViT-L-14s) on multiple datasets. Figure 3 shows that in the specific combination of large models and image corruptions, AutoCLIP does not bring benefits compared to baseline zero-shot classifiers. But still, for 3 out of the 5 VLMs in Figure 3 (and averaged across models), AutoCLIP is better than the baseline zero-shot classifier.
>
> > It is highly suggested to show evidence for "This is inspired by the observation that class descriptors having higher similarity in the embedding space describe the image better". This claim support the rationale of the proposed AutoCLIP. To my understanding, if this claim does not hold, then AutoCLIP does not hold.
>
> We provide evidence for this in the paper in Figure 6 and in Figure 7 (in the appendix): Figure 6 shows that on images from the Food dataset, prompts starting with “A photo of...” get higher weights by AutoCLIP than prompts starting with "Art of ...", while on ImageNetR in Figure 7, it is the other way around. This corresponds to the Food dataset containing actual photos while ImageNetR containing more artistic renditions etc. Thus higher weights correspond to better class descriptor-image fit.
>
> > Is there any speed comparison? How much more inference time does the AutoCLIP need?
>
> We provide some numbers for the case of a ViT-L-14 on the Oxford Pets dataset. Here, encoding an image takes 12.64ms on a V100 (minimum over 100 images). The baseline "averaging" zero-shot classifiers takes additional 0.08ms (average over 640 samples) on top to classify a sample. AutoCLIP takes additional 1.54ms (average over 640 samples) for classification when running bisection for autotuning the step size. For a fixed step size, the overhead of AutoCLIP is only 0.45ms. Thus, AutoCLIP raises inference from 12.64ms to a bit more than 14ms. In contrast, TPT (Shu et al., 2022) and  RLCF (Zhao et al., 2023), which did not report compute or memory requirements, require encoding multiple  image augmentations. TPT states "We augment a single test image 63 times using random resized crops and construct a batch of 64 images, including the original one.", which means that the image encoding time (for instance the 12.64ms from above) is increased by a factor of 64x, plus additional overhead for backpropagating through the text encoder, which likely brings the inference time per sample close to 1s. RLCF also requires multiple image augmentation but does not state the number of augmentations used. In contrast, AutoCLIP does not require any image augmentations or backpropagating through the text encoder and increase inference time from 12.64ms to ~14.0ms instead of ~1 second.

---

### Official Review · Reviewer_5SeE · 2023-11-01

**Soundness:** 3 good
**Presentation:** 3 good
**Contribution:** 2 fair
**Rating:** 6
**Confidence:** 4

**Summary:**

Visual language models such as CLIP are used for zero-shot image classification tasks. This is done by first extracting N text features for a class using N different text prompts, and the cosine similarity between the image feature and the averaged text feature of a class is computed. However, prior works simply averaged over N text features without considering the importance of each text feature. Unlike previous works, this paper proposed to perform a weighted average on the text features during test time by using the cosine similarity between each text feature and the image feature (i.e., higher cosine similarity, higher weight). Extensive experiments and ablations show the effectiveness of the proposed method.

**Strengths:**

1. This paper proposed a simple and effective method for increasing the model performance.
2. Extensive experiments on multiple datasets are conducted and highlight the performance of the proposed method.

**Weaknesses:**

1. The main concern of this work is its novelty. The idea of weighing the text features from different prompts has been proposed [a]. It is unclear why is the proposed method different than [a].
2. While adopting the weighting strategy in [a] during test time is a solution for boosting the classification performance, one could simply average the topK text prompts that have the topK cosine similarity with the image feature. Is the proposed method better?
3. The author is suggested to put the argument explanation of scipy.optimize.bisect to the appendix
4. The x-axis of Figure 2 is not clear. Why are there so many lines on the x-axis? Are they meaningful?
5. Typo: from CLIP (Radford et al., 2021) “we use the” ResNet-50 (RN50) (He et al., 2015), and vision transformers
6. Unclear: “from ∆ = 0.06 for K = 4 over ∆ = 0.33 for K = 10 and ∆ = 0.49 for K = 50 to ∆ = 0.57 for K = 200”
7. Caption of Figure 5 : Shown is Accuracy ….

[a] A Simple Zero-shot Prompt Weighting Technique to Improve Prompt Ensembling in Text-Image Models, ICML2023

**Questions:**

1. The author is suggested to explain the fundamental differences between this work and [a].
2. The author is suggested to show that the proposed method is better than averaging the topK text prompts that have the topK cosine similarity. When K=1, this is equivalent to ``max” aggregation and when K=80 (e.g., for CLIP), which means all the text prompts are averaged. Please ablate the performance for different K.

[a] A Simple Zero-shot Prompt Weighting Technique to Improve Prompt Ensembling in Text-Image Models, ICML2023

---

> ### Author Response · Authors · 2023-11-15
> **Response to Reviewer 5SeE**
>
> We thank the reviewer for the constructive review and helpful feedback. We appreciate that the reviewer considers the proposed AutoCLIP a "simple and effective method for increasing the model performance." and highlights that "Extensive experiments on multiple datasets [...] highlight the performance of the proposed method.". We provide responses to the reviewer's questions and critical remarks below:
>
> > The main concern of this work is its novelty. The idea of weighing the text features from different prompts has been proposed [a]. It is unclear why is the proposed method different than [a].
>
> We thank the reviewer for bringing the work [a] to our attention; we will include a discussion of it in the related works in the next revision of our paper. We note that ZPE (the method from [a]) has more restrictive prerequisites than our AutoCLIP: The "max logit scoring" of ZPE determines prompt weights based on logit statistics in the target domain, which must be computed from multiple images from the target domain. This limits ZPE to operate only with a batch of images instead of a single image. Additionally, the "logit normalization" of ZPE requires availability of image features which represent the feature distribution in the pre-training domain, that is: ZPE with logit normalization is not "source-free". In contrast, our AutoCLIP requires no additional statistics so that we can do the inference with AutoCLIP in a source-free setting from a single input image alone. We note that ZPE and AutoCLIP are orthogonal and could be combined: first normalize logits with ZPE and then construct a auto-tuned zero-shot classifier with AutoCLIP on top.
>
> > While adopting the weighting strategy in [a] during test time is a solution for boosting the classification performance, one could simply average the topK text prompts that have the topK cosine similarity with the image feature. Is the proposed method better?
>
> Thanks for proposing TopR as a baseline (we denote the number of selected queries by R to avoid confusion with the number of total prompt templates that we denote by K in the paper). It is indeed a stronger baseline than max or mean aggregation for suitably chosen R (we note that choosing R is non-trivial in a zero-shot setting). We have run an additional study comparing AutoCLIP against this TopR-CLIP for $100$ DCLIP prompt template, across the same VLMs and datasets as in Figure 2. We provide results for different choices of R, both in terms of the percentage of settings in which AutoCLIP is better as well as the Δ Accuracy compared to TopR-CLIP below.
>
> | TopR | AutoCLIP better | Δ Accuracy
> | ----------- | ----------- | ----------- |
> |  1 | 100% | 5.56 |
> |  5 | 100% | 1.61 |
> | 10 | 92% | 0.79 |
> | 20 | 86% | 0.40 |
> | 50 | 89% | 0.27 |
> |100 | 94% | 0.51 |
>
> We note that even for the optimal choice of R (R=20), which would require tuning R on labeled data, AutoCLIP is better than TopR-CLIP in 86\% of the cases. We will include a more detailed visualization of this study in the appendix of the revised version of the paper.
>
> > The author is suggested to put the argument explanation of scipy.optimize.bisect to the appendix
>
> Thanks for the suggestion, we will do so.
>
> > The x-axis of Figure 2 is not clear. Why are there so many lines on the x-axis? Are they meaningful?
>
> These are just the default ticks of matplotlib for a logarithmic x-axis. We did evaluate for $K \in$ {2, 4, 10, 20, 50, 100, 200, 400, 1000} prompt templates in this figure.
>
> > Unclear: “from ∆ = 0.06 for K = 4 over ∆ = 0.33 for K = 10 and ∆ = 0.49 for K = 50 to ∆ = 0.57 for K = 200”
>
> Here K denotes the number of prompt templates sampled from CLIP/DCLIP/WaffleCLIP and ∆ the average increase in accuracy when using AutoCLIP compared to the baseline "averaging" zero-shot classifier. Did this explanation clarify the statement?

---

> ### Comment · Reviewer_5SeE · 2023-12-04
> **Thanks for the rebuttal**
>
> Thanks to the authors for providing the rebuttal. I carefully checked all the reviews and rebuttals. The authors clarified the novelty of the proposed method compared to [a] and provided additional experiments on the suggested TopR baseline. As shown in Figure 9, AutoCLIP generally performs better than this baseline. This addresses my concerns, and I am prone to increase my rating after the rebuttal.

---

### Official Review · Reviewer_2EvZ · 2023-11-03

**Soundness:** 2 fair
**Presentation:** 3 good
**Contribution:** 2 fair
**Rating:** 5
**Confidence:** 2

**Summary:**

The authors introduce a method called AUTOCLIP that constructs zero-shot classifiers from vision-language models based on considering the statistics of class descriptor-image similarities and automatically adjusting the weights for the templates from prompts. In addition, they discuss how to automatically tune the proposed method's step size that can control the entropy of the prompt template’s weights. The authors evaluate the proposed method AUTOCLIP on several datasets and vision-language models with prompt templates and show the promising results.

**Strengths:**

This paper has good writing and easy to understand and follow the proposed idea. The motivation is reasonable by leveraging the knowledge of visual-language models (VLMs) and automatically tuning the per-image weights to each prompt template at inference time. In addition, they also discuss automatically tuning AUTOCLIP’s step size to control the entropy of the prompt template’s weights. Overall the proposed method is simple with only modifying a few steps in a general zero-shot classifier algorithm. The behind the intuition is also clear and provides significant progress on the zero-shot image classification.

**Weaknesses:**

Even though the proposed method is simple and effective, the method looks like a naive modification method to replace the uniform-weighted average descriptor encodings with weighted average encodings based on the existing algorithms which may limit the paper's novelty.

**Questions:**

1. I'd like to know if the order of the queries is different, will the performance remain the same? If each query has different classes in the inference time, will the weighted average encodings need to be changed/calculated all the time?
2. Similarly, will the proposed method still have the same speed in the inference process or it will be dramatically different?
3. Since the authors claim that the proposed method has the benefit of significantly lowers the test-time computation and memory overhead, it'll be more convincing if adding the comparison with tables, and figures, etc.
4. Is there more discussions about why 'mean' in Figure 5 on Oxford dataset has the worst result?
5. Minor: with the logsumexp providing the best result compared to entropy, mean, and max, is the idea similar to focal loss with multi-class?

---

> ### Author Response · Authors · 2023-11-15
> **Response to Reviewer 2EvZ**
>
> We thank the reviewer for the constructive review and helpful feedback. We appreciate that the reviewer finds our work has "good writing and [is] easy to understand", our method having "intuition [that] is also clear and provides significant progress on the zero-shot image classification". We  provide responses to the reviewer's questions and critical remarks below:
>
> > Even though the proposed method is simple and effective, the method looks like a naive modification method to replace the uniform-weighted average descriptor encodings with weighted average encodings based on the existing algorithms which may limit the paper's novelty.
>
> We agree that AutoCLIP is simple and effective; however, we do not think AutoCLIP is naive since it is a principled solution (gradient-based maximization of an objective) based on a clear intuition: prompt templates resulting in class descriptors more similar to the image on average should get higher weights. Being a simple method is in our opinion a feature not a bug of our contribution.
>
> > I'd like to know if the order of the queries is different, will the performance remain the same?
>
> AutoCLIP is by design permutation-invariant, that is: it treats queries as a set and order of queries does not matter. We also run an empirical sanity check and randomly permuting queries did not change performance.
>
> >  If each query has different classes in the inference time, will the weighted average encodings need to be changed/calculated all the time?
>
> We interpret the reviewer's question that the reviewer likes to know whether the weights can be reused. In fact, the weights in AutoCLIP are recalculated for every sample during inference. If queries or classes change at inference time, the main overhead would be that the class descriptors need to be re-encoded by the VLM's text encoder. This, however, is the case for standard CLIP zero-shot classifiers as well and thus no overhead specific to AutoCLIP.
>
> > Similarly, will the proposed method still have the same speed in the inference process or it will be dramatically different?
>
> AutoCLIP zero-shot classifiers have overhead at inference time compared to standard zero-shot classifiers that simply average encoded class descriptors. However, this overhead is still very small compared to the encoding of the image with the VLM's image encoder in most cases. For instance, in the case of a ViT-L-14 on the Oxford Pets dataset, encoding an image takes 12.64ms on a V100 (minimum over 100 images). The baseline "averaging" zero-shot classifiers takes additional 0.08ms (average over 640 samples) on top to classify a sample. AutoCLIP takes additional 1.54ms (average over 640 samples) for classification when running bisection for autotuning the step size. For a fixed step size, the overhead of AutoCLIP is only 0.45ms. We note that AutoCLIP's overhead is very small compared to other VLM test-time adaption approaches (see below).
>
> > Since the authors claim that the proposed method has the benefit of significantly lowers the test-time computation and memory overhead, it'll be more convincing if adding the comparison with tables, and figures, etc.
>
> Unfortunately, TPT (Shu et al., 2022) and  RLCF (Zhao et al., 2023) did not report compute or memory requirements. However, TPT states "We augment a single test image 63 times using random resized crops and construct a batch of 64 images, including the original one.", which means that the image encoding time (for instance the 12.64ms from above) is increased by a factor of 64x, plus additional overhead for backpropagating through the text encoder, which likely brings the inference time per sample close to 1s. RLCF also requires multiple image augmentation but does not state the number of augmentations used. In contrast, AutoCLIP does not require any image augmentations or backpropagating through the text encoder and increase inference time from 12.64ms to roughly 14ms instead of the other method's ~1 second (for the setting discussed above)
>
> > Is there more discussions about why 'mean' in Figure 5 on Oxford dataset has the worst result?
>
> In general, Oxford Pets is the dataset with the strongest differences between methods (see also y-scale in Figure 2). Why  "mean" performs particularly poorly in this setting is unclear.
>
> > Minor: with the logsumexp providing the best result compared to entropy, mean, and max, is the idea similar to focal loss with multi-class?
>
> Relating AutoCLIP's logsumexp objective function to other losses is interesting. We do not see an immediate connection to focal loss (maybe the reviewer can provide further details on the possible relationship?). However, as stated in the paper we note that −logsumexp has been interpreted as the energy function of a data point (for appropriately trained classifiers) (Grathwohl et al., 2020); in this view, AutoCLIP can be interpreted as minimizing the energy and maximizing the probability density p(x) of x under the zero-shot classifier.

---

### Author Response · Authors · 2023-11-20
**Revision of our Paper**

Dear reviewers and ACs,

We thank you all for the detailed reviews and constructive feedbacks. We have uploaded a revision of our paper that incorporates the reviewer's suggestions and discussion items from our responses. The main changes are (please see the discussion with individual reviewers below for details):
 * As suggested by Reviewer *2EvZ* and *rfek*, we added a discussion of inference-time overhead of AutoCLIP in Section A.1. AutoCLIP raises inference time in a typical setting from 12.64ms to 14.18ms, which is much less than full test-time prompt tuning approaches.
 * We cite and discuss the closely related work by Allingham et al. (2023) in the introduction and related works. Thanks to Reviewer *5SeE* for bringing this work to our attention. The main advantage of AutoCLIP compared to this work is that we adjust weights on single samples (and not on batches) and do not require access to the pre-training feature distribution.
 * We compare to the TopR baseline (suggested by Reviewer *5SeE*) in Figure 9. While TopR aggregation is a stronger baseline than Mean aggregation, AutoCLIP dominates it in the majority of cases (86%) with an average improvement of  0.40 percent point accuracy.
 * Other suggestions by the reviewers such as motivating the zero-shot setting (Reviewer *rfek*), permutation invariance of AutoCLIP wrt. prompt templates (Reviewer *2EvZ*), and minor restructuring and rephrasing (Reviewer *5SeE*) have also been incorporated.

We would greatly appreciate if the reviewers would revise their reviews based on our revised document and our responses. We also appreciate any further comments, questions, or suggestions.

---

### Meta-Review · Area_Chair_KAf5 · 2023-12-06

**Metareview:**

This paper proposes AutoCLIP, a method for auto-tuning zero-shot classifiers for vision-language models like CLIP. Three reviewers reviewed the paper. The final ratings were 5, 6, 6. Positive points include the promising approach, extensive experimental results, and clear presentation.  Some common concerns include incremental novelty, missing baselines, question on significance of marginal improvements, and limitation to the zero-shot setting. The rebuttal partly addressed some of these concerns regarding missing baselines, and to some degree incremental novelty.  However, the limitation to the zero-shot setting and significance of the marginal improvement in results were not adequately addressed (the corresponding reviewer's comments are copied below in case they are not visible to the authors). Overall, this is a borderline paper with split reviews and none of the reviewers strongly supportive of acceptance. The paper, rebuttal, discussion, and author messages were carefully discussed among the ACs, and the ACs agree that the paper has limited novelty and the experimental results are not strong enough to support acceptance.  The ACs would like to encourage the authors to improve the paper and resubmit to another conference.

=====
From Reviewer rfek:
Thanks authors for the responses to my questions. I have read authors' responses and other reviewers' comments. Part of my concerns have been addressed by the authors responses. However, my first and second concerns regarding the method significance were not well addressed. The authors only said "zero-shot classification is important enough, 1% improvement is significant", but did not explain why. "zero-shot classification is important enough, 1% improvement is significant" is not informative. The authors did not explain what this 1% improvement can bring. I will keep my original rating in case some concerns are addressed while some are not.

**Justification For Why Not Higher Score:**

The paper is not ready for acceptance based on the points written in the meta review.

**Justification For Why Not Lower Score:**

N/A

---

### Decision · Program_Chairs · 2024-01-16

Reject